# Transcriptomics-Based Drug Repurposing Approach Identifies Novel Drugs against Sorafenib-Resistant Hepatocellular Carcinoma

**DOI:** 10.3390/cancers12102730

**Published:** 2020-09-23

**Authors:** Kelly Regan-Fendt, Ding Li, Ryan Reyes, Lianbo Yu, Nissar A. Wani, Peng Hu, Samson T. Jacob, Kalpana Ghoshal, Philip R.O. Payne, Tasneem Motiwala

**Affiliations:** 1Department of Biomedical Informatics, The Ohio State University College of Medicine, Columbus, OH 43210, USA; kelly.regan@osumc.edu (K.R.-F.); lianbo.yu@osumc.edu (L.Y.); 2Department of Pathology, The Ohio State University College of Medicine, Columbus, OH 43210, USA; ding.li@osumc.edu (D.L.); peng.hu@osumc.edu (P.H.); Kalpana.ghoshal@osumc.edu (K.G.); 3Comprehensive Cancer Center, The Ohio State University, Columbus, OH 43210, USA; 4Department of Cancer Biology and Genetics, The Ohio State University College of Medicine, Columbus, OH 43210, USA; ryan.reyes@cardinalhealth.com (R.R.); samson.jacob@osumc.edu (S.T.J.); 5Department of Biotechnology, School of Life Sciences, Central University of Kashmir, Ganderbal, Jammu and Kashmir 191201, India; waninh@cukashmir.ac.in; 6Washington University Institute for Informatics, Washington University in St. Louis School of Medicine, St. Louis, MO 63108, USA; prpayne@wustl.edu

**Keywords:** hepatocellular carcinoma, drug resistance, drug repurposing, sorafenib, fostamatinib, dasatinib

## Abstract

**Simple Summary:**

Hepatocellular carcinoma (HCC), a type of liver cancer, remains a treatment challenge due to late detection and resistance to currently approved drugs. It takes 15–20 years for a single new drug to become FDA approved. The purpose of this study was to expedite identification of novel drugs against drug-resistant HCC. For this, we matched gene expression alterations in resistant HCC with gene expression changes caused by treatment of cancer cells with drugs already FDA approved for other diseases to find the drug that can reverse the resistance-related changes. Among the identified drugs, we validated the growth inhibitory effect of two drugs, identified their mechanism in HCC and, thus, provided proof of concept evidence for validity of this drug repurposing approach with potential for use in personalized medicine.

**Abstract:**

*Objective:* Hepatocellular carcinoma (HCC) is frequently diagnosed in patients with late-stage disease who are ineligible for curative surgical therapies. The majority of patients become resistant to sorafenib, the only approved first-line therapy for advanced cancer, underscoring the need for newer, more effective drugs. The purpose of this study is to expedite identification of novel drugs against sorafenib resistant (SR)-HCC. *Methods:* We employed a transcriptomics-based drug repurposing method termed connectivity mapping using gene signatures from in vitro-derived SR Huh7 HCC cells. For proof of concept validation, we focused on drugs that were FDA-approved or under clinical investigation and prioritized two anti-neoplastic agents (dasatinib and fostamatinib) with targets associated with HCC. We also prospectively validated predicted gene expression changes in drug-treated SR Huh7 cells as well as identified and validated the targets of Fostamatinib in HCC. *Results:* Dasatinib specifically reduced the viability of SR-HCC cells that correlated with up-regulated activity of SRC family kinases, its targets, in our SR-HCC model. However, fostamatinib was able to inhibit both parental and SR HCC cells in vitro and in xenograft models. Ingenuity pathway analysis of fostamatinib gene expression signature from LINCS predicted JAK/STAT, PI3K/AKT, ERK/MAPK pathways as potential targets of fostamatinib that were validated by Western blot analysis. Fostamatinib treatment reversed the expression of genes that were deregulated in SR HCC. *Conclusion:* We provide proof of concept evidence for the validity of this drug repurposing approach for SR-HCC with implications for personalized medicine.

## 1. Introduction

Liver cancer ranks seventh among the most common cancers in the world and, according to a recent report [1], it is the fourth leading cause of cancer-related death. As per the prediction of The American Cancer Society, ~42,810 new individuals (30,170 in men and 12,640 in women) will be diagnosed with primary hepatocellular cancer and intrahepatic bile duct cancer and about 30,160 patients (20,020 men and 10,140 women) will die of these cancers in the United States in 2020 (https://www.cancer.org/cancer/liver-cancer.html). Since 1980, liver cancer incidence rates have more than tripled and death rates have more than doubled.

While a small proportion of hepatocellular carcinoma (HCC) patients diagnosed at an early stage can be treated by tumor resection, cryoablation or liver transplant, these treatments are not effective in the majority of HCC patients diagnosed at an advanced stage of the disease. Sorafenib, a multi-kinase inhibitor, is the most widely used drug since 2007 in treating such patients [2]. However, the median overall survival of sorafenib treated patients is only extended by 2.8 months compared to untreated patients [3]. This minimal therapeutic response is attributed to HCC tumors having an intrinsic resistance to the cytostatic effects of sorafenib [4]. For HCC patients who become resistant to sorafenib, FDA recently approved its congener regorafenib [5] and checkpoint blockade anti-PD-1 antibodies, nivolumab [6] and pembrolizumab [7], as second-line treatment. However, only a subset of such patients respond to this combination therapy. In recent clinical trials in the first-line setting, nivolumab [8] or pembrolizumab [9] could not significantly improve survival of HCC patients compared to sorafenib and best supportive care, respectively. Lenvatinib, another multi-kinase inhibitor, recently approved as first-line therapy for advanced HCCs, was not significantly superior to sorafenib in improving overall survival in clinical trials [10]. Thus, there is an urgent need to develop therapeutic strategies to overcome sorafenib resistance and discover new, more effective therapies.

Due to the heterogeneous molecular mechanisms underlying HCC tumor progression and sorafenib resistance, it is unlikely that targeting one major molecular mechanism will be sufficient to treat this disease [11]. Furthermore, collecting tissues through biopsy is not standard of care for advanced HCC patients that relapse on sorafenib. As a result, the lack of available biomolecular data of HCC patients treated with sorafenib severely limits the ability to study fundamental mechanisms of resistance and potential targets for combination therapies. Therefore, we sought to investigate sorafenib resistance in HCC in an unbiased way through global analysis of the transcriptome in experimental models of sorafenib resistance. We generated gene expression data from an in vitro model of HCC sorafenib resistance in the Huh7 cell line, and conducted a comprehensive analysis of other publicly available gene expression data from experimental models of sorafenib resistance (SR-HCC) and patient-derived HCC tumors. We evaluated these SR-HCC gene expression models for their coverage in human HCC tissue samples and their prognostic significance. Next, in order to discover drug-disease hypotheses, we utilized the aforementioned gene expression profiles as the basis for our computational drug repurposing analyses via connectivity mapping. Connectivity mapping uses pattern-matching algorithms to compare genome-wide gene expression changes observed in cultured human cells treated with drugs to those of biological states of interest: e.g., tumor vs. normal [12]. Connectivity scores quantify the drug-disease hypotheses through correlations between ranked gene lists of query gene signatures and drug reference gene signatures, commonly via the Kolgomorov–Smirnov statistic or the modified gene set enrichment analysis method [12,13]. For instance, drug-induced gene signatures with negative connectivity scores are hypothesized to reverse or oppose the query gene signature characterizing a disease, and vice versa. Furthermore, the use of genome-wide expression profiles provides mechanistic insight into tumor biology and drug efficacy, which may be missed by other guilt-by-association approaches. In this study, we hypothesize that transcriptomics data from experimental models of sorafenib-resistant HCC (i) will enable validation of the in vitro models in the absence of tissue available from sorafenib resistant tumors, (ii) can be applied in connectivity mapping studies to predict novel therapies to curb resistance to sorafenib in HCC, and (iii) will reveal molecular mechanisms underlying sorafenib resistance in HCC tumors in the context of gene targets and enriched pathways.

## 2. Results

A workflow overview for this study is presented in Figure 1.

### 2.1. Comparison of Gene Signatures of Experimental Models of Sorafenib Resistance

#### 2.1.1. Generation and Microarray Analysis of Sorafenib-Resistant HCC Cell Line

Sorafenib-resistant HCC cell lines were generated from parental (sensitive) Huh7 cells (Huh7-S) following long-term exposure to sorafenib [14]. Viability assays demonstrated that the IC_50_ doses for resistant cells were approximately 5-fold higher than that of parental Huh-7 cells (Appendix A). Microarray analysis was performed on the parental cells (Huh7-S) and sorafenib-resistant cells (Huh7-R and Huh7-R-A7) (GSE94550). Comparing Huh7-R-A7 vs. Huh-S cells to define an HCC sorafenib resistance gene signature, we determined 368 genes to be differentially expressed (adjusted *p* < 0.001) (Appendix A). The top Gene Ontology (GO) molecular functions, biological processes and cellular components enriched in the sorafenib resistance gene signature (FDR < 0.05) are reported in Appendix A.

#### 2.1.2. Evaluation of Experimental Models of HCC Sorafenib Resistance against HCC Patient Datasets

We conducted a comprehensive analysis of publicly available gene expression datasets characterizing sorafenib resistance in HCC, including an in vitro HepG2 cell line (HepG2-R) [15], in vitro HCC patient culture (HCC-3sp-R) [16], and an in vivo xenograft model using transplanted Huh7 cells (Xeno-R) [17]. Sorafenib resistance (SR) dataset features are described in Table 1, and gene signature overlap is shown in Appendix A. Interestingly, the distinct gene signatures showed very little overlap among up- and down-regulated genes. To assess the clinical prognostic relevance of each of the SR gene signatures, we obtained additional publicly available gene expression datasets characterizing HCC patient tumors of diverse etiologies from The Cancer Genome Atlas (TCGA) and Gene Expression Omnibus (GEO) databases. First, the presence of the SR gene signatures within the HCC patient datasets was detected using the nearest template method (weighted cosine similarity metric, FDR < 0.05). The ability for the gene signatures to distinguish between HCC tumor and normal liver tissue was comparable across datasets. However, the average frequency of SR+ HCC tumors defined by the SR gene signatures among the GEO datasets were highest in the Huh7-R-A7 (52.7%) and Xeno-R (46.4%) signatures, while HepG2-R (26.3%) and HCC-3sp-R (27.7%) were lower (Table 1). Similarly, the Huh7-R-A7 and Xeno-R gene signatures detected a higher percentage of SR+ samples using RNAseq data from the TCGA liver hepatocellular carcinoma (LIHC) subset (Figure 2A). Second, we assessed the relationship between the presence of a sorafenib resistance gene signature and survival using the TCGA dataset. Of the four SR gene signatures, HCC patients with tumors harboring the Huh7-R-A7 sorafenib resistance gene signature (SR+) exhibited significantly reduced survival as compared to those that showed the opposite gene signature pattern (SR−) (log-rank *p* = 0.0086; HR = 1.59, 95% CI: 1.13–2.45) (Figure 2B). Due to the lack of survival outcome data for HCC patients with gene expression data from the GEO database, we assessed the sensitivity and specificity by which the four SR gene signatures could distinguish HCC tumor vs. normal liver tissue. The average sensitivity and specificity for the SR signatures are as follows: Huh7-R-A7 (0.50, 0.82), HepG2-R (0.53, 0.89), HCC-3sp-R (0.48, 0.89), Xeno-R (0.73, 0.80).

### 2.2. Drug Repurposing Predictions to Reverse Sorafenib Resistance in HCC

#### 2.2.1. Library of Integrated Network-Based Cellular Signatures (LINCS) Analysis for Drug Repurposing Predictions

To identify drugs that can reverse sorafenib resistance in HCC, connectivity mapping analyses were conducted via the Library of Integrated Network-based Cellular Signatures (LINCS) L1000 system. This database consists of 476,251 gene expression profiles of drug and genetic perturbation conditions across 77 cellular contexts. We chose to focus on drug-treated profiles from the HepG2 cell line, as it represented the HCC cell line with the greatest number of unique drugs (n = 3740). To determine whether HCC cell line context can affect the distribution of drug repurposing hypotheses, we first used gene expression profiles for 18 HCC cell lines from the CellMinerHCC database to query HepG2 drug perturbation gene expression profiles from the LINCS database. We then performed hierarchical clustering analysis of connectivity scores for drug predictions across the 18 HCC cell lines. The heatmap shown in Figure 2C revealed two distinct clusters of drug connectivity scores. Interestingly, both HepG2 and Huh7 belonged to the same main cluster branch, suggesting that the LINCS HepG2 represents a suitable system to generate drug predictions from gene expression profiles originating in Huh7 cell line models. Next, we compared the connectivity scores across the four HCC sorafenib resistance models via hierarchical clustering analysis (Figure 2D). We observed that drug prediction patterns derived from the two cell lines (Huh7-R-A7 and HepG2-R) and short-term culture (HCC-3sp-R) were nearly opposite those derived from the mouse model (Xeno-R). The highest degree of similarity of drug prediction patterns was between HCC-3sp-R and HepG2-R. These results are consistent with the observed gene signature overlap shown in Appendix A. For instance, the maximum observed gene overlap between HCC-3sp-R and HepG2-R was 88 up-regulated and 189 down-regulated genes, while Xeno-R showed the least overlap with any other model.

#### 2.2.2. Drug Target Analysis and Candidate Prioritization

Due to the superiority of the Huh7-R-A7 sorafenib resistance gene signature to distinguish significant patient survival patterns using the TCGA LIHC data (Figure 2B), we focused our prioritization and validation efforts for LINCS drug predictions from this gene signature. Amongst the drugs predicted to reverse sorafenib resistance from our LINCS analyses, we first determined the approval and investigational status from the DrugBank and Aggregate Analysis of ClinicalTrials.gov (AACT) databases to prioritize those drugs that would be most feasible for future preclinical and clinical testing (Appendix A). For these drugs, we annotated drug activity, target pathway and gene information from KEGG and DrugBank databases. We then further selected only those drugs with known anti-neoplastic activity for initial validation. Finally, we conducted a systematic search of genes associated with hepatocellular carcinoma using a publicly available literature mining tool [18] to determine whether the genes targeted by drug candidates had known roles in HCC. The final prioritized drug list is shown in Table 2. We selected two drug candidates, one representative from each approval status, for subsequent validation: 1) dasatinib, SRC family of kinases inhibitor (FDA-approved) and 2) fostamatinib, SYK inhibitor (under clinical investigation). Additionally, we analyzed a protein-protein interaction (PPI) network of drug target genes, as shown in Appendix A. The initial drug targets were connected in a network of a total of 322 protein nodes through 1029 total interactions (edges). The average node degree (interaction partners) is 6.39, and the number of observed edges is significantly higher than expected (n = 475; PPI enriched *p* value < 0.05). PPI connections were recovered for five out of seven dasatinib gene targets, including SRC. A community detection network algorithm was applied to the network, and both SRC and SYK were found in the largest PPI module 5 (n = 46 total nodes). Notably, SRC was ranked with the third highest node degree (n = 33 connections), while SYK has n = 8 connections. SYK exhibited a clustering coefficient of 0.57, while SRC has a clustering coefficient of 0.14. Taken together, these results suggest SRC inhibition may have a broad impact on the overall network, while SYK appears to be involved with a more tightly regulated group of proteins. Finally, we selected the top 100 central genes based on the eigencentrality measure, which included all dasatinib and fostamatinib gene targets that were in the network. We found that these highly central drug target genes were altered (mutations, copy number variations, mRNA and protein expression levels) at a higher frequency in the SR+ TCGA patients (58%) as compared to the SR- patients (31%). The type and frequency of gene alterations of the top central genes in the PPI network in SR+ and SR- TCGA LIHC patient tumors are shown in Appendix A.

#### 2.2.3. Validation of SRC-Inhibitor, Dasatinib, and SYK-Inhibitor, Fostamatinib, Alone and in Combination with Sorafenib

Dasatinib and fostamatinib were initially tested in vitro as single agents in HCC cell lines (parental Huh7-S, resistant pool Huh7-R and resistant clone Huh7-R-A7). Parental and sorafenib-resistant Huh7 cells were treated with increasing concentrations of dasatinib and fostamatinib independently, and cellular viability was assessed after 48 h. Sorafenib-resistant Huh7 cells were significantly more sensitive to dasatinib toxicity than parental cells (Figure 3A). Parental cells displayed an IC_50_ of > 60 µM, while the IC_50_ of resistant cells was < 10 µM. On the contrary, all three cell lines displayed similar sensitivity to fostamatinib toxicity with IC_50_ values between 20 and 35 µM (Figure 3B), indicating that fostamatinib may be useful both before and after resistance to sorafenib occurs. We next hypothesized that combining either dasatinib or fostamatinib with sorafenib would synergistically inhibit cell viability.

We thus assessed the effect of dasatinib and fostamatinib treatment, alone and in combination with sorafenib, on the reproductive ability of single cells using the clonogenic survival assay (Figure 3C,D). At the low (2 µM) concentration tested, dasatinib completely inhibited colony formation in Huh7-R cells, while having little effect on parental cells. This observation is consistent with the higher sensitivity of resistant cells to dasatinib observed in viability assay (Figure 3A). Furthermore, the addition of sorafenib did not further sensitize the Huh7-S cells to dasatinib. Similar to the viability assay, both parental and resistant cells were significantly sensitive to fostamatinib. The combination of fostamatinib and sorafenib appeared to have a greater ability to inhibit colony formation in Huh7-S, Huh7-R and Huh7-R-A7 cells compared to either drug alone.

Since sorafenib-resistant HCC cells appeared to be much more sensitive to long-term dasatinib toxicity than non-resistant HCC cells, we hypothesized that the activity of SRC kinases would be up-regulated in the resistant cells. Kinase array analysis demonstrated that five out of seven SRC kinases (Src, Yes, Fyn, Lck and Lyn) were significantly hyper-phosphorylated in resistant cells as compared to parental cells (Figure 4A and Appendix A). On the contrary, SYK protein, the target of fostamatinb was not detected in the Huh7 cells. Therefore, in order to delineate possible mechanisms for the action of fostamatinib, we obtained the gene expression characterizing fostamatinib-treated HepG2 cells (10 μM, 6 h post-treatment) from the LINCS database, and performed Ingenuity pathway analysis. Several pathways were identified as significantly deregulated post-treatment: inhibition of oncogenic signaling through JAK/STAT (*p* < 1.0 × 10^−8^), protein kinase A (*p* < 1.0 × 10^−7^), PI3K/AKT (*p* < 1.0 × 10^−4^), STAT3 (*p* < 1.0 × 10^−4^) and ERK/MAPK (*p* < 1.0 × 10^−3^); and activation of the tumor suppressor PTEN (Figure 4B).

#### 2.2.4. Characterization of Fostamatinib as Anti-HCC Drug

Fostamatinib, recently FDA approved for immune thrombocytopenia, has been clinically tested for autoimmune diseases and lymphoma. However, its anti-HCC efficacy has not been explored previously. Since fostamatinib could inhibit the growth of both sorafenib-sensitive and -resistant cells, we tested its inhibitory potential in several HCC cell lines with differing sensitivities to sorafenib. The IC50 of fostamatinib for most of these cell lines was 2–4 uM (Appendix A). To confirm the in vivo efficacy of fostamatinib, we used a subcutaneous xenograft model of MHCCLM3 cells that inherently have reduced sensitivity to sorafenib. The growth of tumors was significantly inhibited in fostamatinib-treated mice compared to vehicle-treated mice (Figure 5A). Comparable body weights of the tumor-bearing mice treated with vehicle or fostamatinib (Figure 5A) indicate no systemic toxicity of the drug. Tumors harvested at the end of 28 days of treatment weighed an average of 3.2 g in vehicle-treated mice and 1.8 g in fostamatinib-treated mice (Figure 5A). However, due to the small sample size, this difference was not statistically significant (*p* = 0.1031).

Based on the IPA analyses of the gene expression profile of fostamatinib-treated HepG2 cells from LINCS (Figure 4B), we performed immunoblot analysis to validate the kinases that are inhibited by fostamatinib. Our data demonstrate that fostamatinib inhibits phosphorylation of ERK, AKT and STAT3 but has no effect on PTEN. Additionally, we observed inhibitory effect on EGFR and JNK (Figure 5B).

Finally, we sought to validate predicted gene expression changes associated with fostamatinib treatment from LINCS in our Huh7 cell line model. We performed RNA-seq analysis of fostamatinib- vs. DMSO vehicle-treated sorafenib-resistant Huh7-R cells and determined the concordance of fold changes for differentially expressed genes (adjusted *p* < 0.05) with both the LINCS fostamatinib gene expression signature and Huh7-R sorafenib resistance gene expression signature (Appendix A). Prioritized differentially expressed genes that were concordant with the LINCS fostamatinib signature and discordant with the Huh7-R sorafenib resistance signature are shown in Appendix A. We observed 18 up-regulated and 9 down-regulated genes significantly differentially expressed genes in the Huh7-R signature that were reversed in the fostamatinib treated Huh7 cells and LINCS fostamatinib HepG2 signature. Several examples of genes implicated in HCC tumorigenesis and/or sorafenib resistance that were up-regulated in the Huh7-R cells and down-regulated following fostamatinib treatment in both Huh7-R and HepG2 HCC cells include transforming growth factor beta 2 (TGFB2) [19,20], hypoxia inducible factor 1 alpha subunit (HIF1A) [21], intercellular adhesion molecule 1 (ICAM1) [22,23] and ETS proto-oncogene 1 transcription factor (ETS1) [24].

### 2.3. Analysis of Clinical and Demographic Factors

#### 2.3.1. Effect of Etiology with Drug Repurposing Hypothesis

We assessed whether HCC etiology could influence drug repurposing predictions. The GEO HCC patient gene expression datasets, described in Table 1, were filtered to include patient tumors specific to a given HCC etiology: hepatitis B virus (HBV), hepatitis C virus (HCV) and alcohol-induced (AI). Etiology-specific gene expression signatures were used in connectivity mapping analysis. Using hierarchical clustering analysis of drug connectivity scores, we found that all three HBV patient datasets clustered together, and that two of the three HCV patient datasets clustered together with the one AI patient dataset (Figure 6A).

#### 2.3.2. Association of Clinical and Demographic Factors with Sorafenib Resistance Signature

We assessed whether clinical factors (etiology, clinical stage, Child–Pugh class) and patient demography (race, gender) influenced sorafenib resistance gene signature status. We observed that HCC etiology could affect the proportion of tumors harboring SR+ and SR− gene signatures in the TCGA LIHC dataset (chi-square test, *p* = 0.0279), as shown in Figure 6B. Two etiologies that exhibited significant differences in proportion of SR+/SR− HCC tumors included HBV (Fisher exact test, *p* = 0.0444) and non-alcoholic fatty liver disease (NAFLD) (Fisher exact text, *p* = 0.0180). We also examined the proportion of SR+ and SR− HCC tumors among different stages, Child-Pugh class, race and gender (Figure 6C–F). HCC patient race exhibited a significant trend (chi-square test, *p* = 0.004), as shown in Figure 6E, where a higher proportion of SR+ and SR- HCC tumors were found in white and Asian patients, respectively. However, no significant trend was observed for clinical stage, Child–Pugh class or gender among SR+ vs. SR− HCC tumor status.

## 3. Discussion

In this paper, we address how different publicly available gene expression datasets derived from in vitro and in vivo models of sorafenib resistance in HCC may be reliably assessed in HCC patient tissue in terms of their prognostic significance and ability to derive drug repurposing predictions. We also provide proof of concept evidence for the successful use of the computational drug repurposing approach in the context of sorafenib-resistant HCC. Similarly, we report on differences in drug repurposing hypotheses with respect to HCC etiology and cell line source. Although connectivity mapping has been applied across diverse domains, this is the first study that has sought to explicitly identify drugs for sorafenib-resistant HCC using drug-induced gene expression signatures from the Library of Integrated Network-based Cellular Signatures (LINCS) database.

Several previous studies have applied HCC gene signatures to the Connectivity Map (CMap) database, which consists of a collection of gene expression profiles of five human cancer (non-HCC) cell lines treated with 1309 compounds [12]. Two studies used HCC gene expression signatures to query against the CMap database, and validated several drug candidates in vitro and in vivo [25,26]. Another group used a combination of CMap and LINCS to discover novel HCC drugs, and validated three anthelminitics in primary hepatocytes and two mouse models [27]. Lv et al. queried CMap using the HCC-3sp-R gene expression data evaluated in this study, and generated six drug predictions to reverse resistance to sorafenib in HCC; however, these predictions were not validated in any context in this publication and no information regarding drug mechanisms was presented [11]. While these previous studies have demonstrated the feasibility of validating connectivity mapping predictions in HCC, all studies to date have been mostly limited to the use of the CMap drug reference database. Two distinct advantages of this LINCS-based study include the ability to gauge the effect of drug perturbations on liver cancer cells (HepG2), as well as the increased number of perturbagens tested in the LINCS systems (n = 3740 total in HepG2 cell line).

In this study, fostmatinib was discovered to be a potentially useful first-line therapy or following resistance to sorafenib, either alone or in combination with sorafenib. Fostamatinib is an inhibitor of spleen tyrosine kinase (SYK), and is currently under investigation for the treatment of several autoimmune diseases [28]. Fostamatinib has been shown to have anti-cancer properties for hematological malignancies [29,30], and this is the first study investigating its use in HCC and sorafenib resistance. SYK is a non-receptor cytoplasmic tyrosine kinase involved in signal transduction in cells of hematopoietic origin, and more recently, implicated both as a tumor suppressor and promoter of cell survival in various hematopoietic and epithelial cancers [31,32]. Reduction in SYK expression has been described as a potential prognostic biomarker in several cancers, including HCC [33,34,35,36]. Although SYK mRNA has prognostic significance in HCC, the lack of its expression at protein level in some HCC cell lines that are sensitive to fostamatinib suggested that the drug functions through other targets that remain to be discovered. We have demonstrated that several kinases (ARK, AKT, STAT3, EGFR, and JNK) are inhibited by fostamatinib but future follow-up studies will be needed to identify direct target(s) of fostamatinib in HCC.

Dasatinib was confirmed to have a unique role in inhibiting cell growth of sorafenib-resistant HCC cells. The Src family kinase inhibitor dasatinib is approved for the treatment of Ph+ chronic myeloid leukemia (CML) in chronic phase and imatinib-resistant disease, Ph+ acute lymphoblastic leukemia with resistance to prior therapy, and is under clinical investigation for solid cancers. Src family kinase activity has been implicated in several oncogenic processes, including cellular proliferation, survival, migration and angiogenesis, and increased activity has been demonstrated in HCC in vitro [37,38,39]. Our experiments demonstrated that dasatinib, a Src family kinase inhibitor, was effective in reducing HCC cell viability and colony formation alone and in combination with sorafenib in sorafenib-resistant HCC cells. Additionally, we showed that Src family kinases were significantly activated in the sorafenib-resistant HCC cells as compared to sorafenib-sensitive HCC cells, consistent with the known mechanism of action of dasatinib.

Recently, dasatinib was shown to be successful in reducing HCC cell proliferation, adhesion, migration and invasion in vitro via inhibiting Src and several downstream signaling pathways, including PI3K/PTEN/Akt and SFK/FAK [40]. Another study found that phosphorylation of Src was inhibited in a panel of HCC cells that were sensitive and resistant to dasatinib, and that cell proliferation was not affected by knocking down Src and p-Src in dasatinib-sensitive cells. The authors concluded that dasatinib-mediated inhibition of Src alone is not sufficient to induce its anti-proliferative or pro-apoptotic effects, and that dasatinib may mediate its effects via other targets in addition to Src [41]. Finally, dasatinib was tested in patients with advanced HCC in a recent phase II clinical trial (NCT00459108), but was terminated early due to futility. The primary objectives were to determine the progression-free survival (PFS) rate and response rate at 4 months in patients with unresectable advanced HCC treated with dasatinib. Several factors that may have influenced the results of this clinical trial include compromised liver status in advanced HCC patients and the use of RECIST criteria to determine response rate, which is known to be ineffective in evaluating cytostatic agents, including sorafenib [42]. Furthermore, our results suggest that dasatinib may be most useful for an enriched HCC patient population with transcriptomic biomarkers characteristic of sorafenib resistance.

We also found that HCC etiology may influence sorafenib resistance and drug repurposing hypothesis generation using the transcriptomics-based LINCS system. Interestingly, sorafenib was previously observed to be more effective for HCC patients with an underlying HCV infection compared to HBV infection or alcoholic cirrhosis [43]. In another study, dasatinib was shown to be most effective in a group of HCC patients with a “progenitor molecular subtype”, as assessed by gene expression profiling [41]. Taken together, these findings highlight the importance of considering HCC patient etiology and other molecular features in drug prediction studies.

## 4. Materials and Methods

### 4.1. Reagents (Drugs and Antibodies)

Dasatinib (S1021) and fostamatinib (S2206) were obtained from Selleckchem (Houston, TX, USA). Sorafenib (S-8502) was purchased from LC Laboratories (Woburn, MA, USA). Antibodies were purchased from either Santa Cruz Biotechnology (Dallas, TX, USA) (EGFR #sc-03; STAT3 #sc-482; β-actin #s-47778) or Cell Signaling Technology (Danvers, MA, USA) (pAkt #4060; Akt #9272; pERK #4370; ERK #9102; pEGFR #3777; pSTAT3 #9131; pPTEN #9551; PTEN #9559; pJNK #9251; JNK #9252)

### 4.2. Cell Culture and Acquired Sorafenib Resistance

All cells were maintained in minimum essential media supplemented with L-glutamine (2 mM), 10% FBS, sodium pyruvate (0.11 g/L) and penicillin/streptomycin (100 U/mL). Cell media for sorafenib resistant cell lines were also supplemented with sorafenib (6 µM and 0.1% DMSO). Sorafenib was withdrawn from the cell media of resistant Huh7 cells for 5–7 days prior to performing all experiments. The HCC cell line Huh7 cells were generously provided by Dr. James Taylor (Fox Chase Center, PA, USA). Sorafenib resistant cells (Huh7-R) and several resistant clones (Appendix A) were generated as described previously [14]. Huh-R-A7 was randomly selected to represent sorafenib resistant clones for all subsequent experiments.

### 4.3. Cell Viability Assay

Cells were seeded into 96-well plates (~2000 cells/well) and allowed to incubate overnight. The next day, cell media were replaced with MEM media containing specified concentration of sorafenib, fostamatinib or dasatinib (with 0.1% final DMSO concentration). After 48 h of treatment, the CellTiter-Glo^®^ luminescent viability assay was utilized following the manufacturer’s instructions. Prior to measuring viability with a luminometer, the luminescent supernatant was transferred to an opaque luminometer 96-well plate.

### 4.4. Colony Formation Assay

Cells were seeded into 6-well plates (~2000–5000 cells/well) and allowed to incubate for 24–48 h. Cells were then treated with a continuous dose of therapeutics (with 0.1% final DMSO concentration) for 14–18 days. Media were replaced every 2–3 days. After colonies grew to a sufficient size, cells were fixed with 3.7% paraformaldehyde (in PBS) and stained with a 0.05% crystal violet solution.

### 4.5. Mouse Strains, Animal Husbandry and Treatment

Male NSG (NOD scid gamma) mice were purchased from Target Validation Shared Resource (TVSR) core facility at the Ohio State University. All animals were housed in a temperature-controlled room under a 12 h light/12 h dark cycle and under helicobacter-free conditions and fed normal chow diet. All animal studies were reviewed and approved by the Ohio State University Institutional Laboratory Animal Care and Use Committee (Protocol # 2008A0236). For drug treatment, fostamatinib disodium (Cat# DC1013, DC-Chemicals, Shanghai, China) was dissolved with 30% polyethylene glycol 400 (PEG 400) (Cat# 91893, Sigma-Aldrich, St. Louis, MO, USA), 5% propylene glycol (PPG) (Cat# P4347, Sigma-Aldrich), and 0.5% Tween 80 (Cat# P4780, Sigma-Aldrich) to the final concentration of 20 mg/mL immediately before delivery through oral gavage. For subcutaneous xenografts, 10–12 weeks old NSG mice were injected subcutaneously with HCCLM3 (2.5 × 10^6^) cells into the right flank. When tumor volume reached 100 mm^3^, mice were randomized into 2 groups for treatment with either vehicle (30% PEG 400, 5% PPG and 0.5% Tween 80) or fostamatinib disodium (80 mg/kg) administrated daily through oral gavage. Tumor volumes based on digital caliper measurements were calculated by the ellipsoidal formula (1/2(length × width^2^)). After 28 days of treatment, mice were euthanized and tumor tissues were collected, weighed and photographed.

### 4.6. Western Blot Analysis

In total, 5 × 10^5^ HCC cells of Huh7 and Huh7-SR (sorafenib resistant) were plated in 60 mm dishes overnight and treated with DMSO and Fostamatinib (10 µM) for 30 min. Whole cell extracts were prepared in cell lysis buffer (Cat# 9803, Cell signaling technology, Beverly, MA) containing protease inhibitor cocktail (#P8340, Sigma-Aldrich, St. Louis, MO, USA) and phosphatase inhibitor cocktails (#P5726 and P0044, Sigma-Aldrich, St. Louis, MO). The cell lysates were incubated at 4 °C for 10 min and centrifuged at 4 °C for 10 min to collect clear supernatants. Protein concentrations in the extracts were measured by the bicinchoninic acid (BCA) method using BSA as the standard. Equivalent amounts of proteins from whole cells were separated by SDS-polyacrylamide (10%) gel electrophoresis (Bio-Rad, Hercules, CA, USA), transferred to nitrocellulose membranes (GE Healthcare, Chicago, IL, USA), and incubated using blocking buffer (LI-COR, Lincoln, NE, USA) followed by immunoblotting with phospho-Akt (S473), total Akt, phospho-ERK (Thr202/Tyr204), total ERK, phospho-EGFR (Y1086), total EGFR, phospho-STAT3 (Y705), total STAT3, phospho-PTEN (Ser380), total PTEN, phospho-JNK (Thr183/Tyr185), total JNK, and β-actin. Catalogue numbers and sources of the antibodies are provided in the Reagents section. Following incubation with appropriate secondary antibody (IRD-680 or IRD-800), the specific immune-reactive bands were visualized using Odyssey CLx Imaging System (LI-COR, Lincoln) and quantified using Image Studio 5.2.5 software (LI-COR, Lincoln, NE, USA).

### 4.7. Human Phospho-Protein Array

Cell lysates of Huh7 parental and Huh7 sorafenib-resistant pool cells were subjected to phosphoprotein analysis using the Proteome Profiler Human Phospho-Kinase Array Kit (#ARY003B, R & D Systems, Minneapolis, MN, USA) following the manufacturer’s protocol to quantify phosphorylation levels of 43 proteins phosphorylated at tyrosine, serine, or threonine residues.

### 4.8. Differential Gene Expression Analysis

#### 4.8.1. Sorafenib Resistance Gene Expression Signatures

Microarray analysis was performed on the parental cells (Huh7-S), a pool of sorafenib-resistant cells (Huh7-R) and sorafenib-resistant clone A7 (Huh7-R-A7) using the Affymetrix GeneChip Human Transcriptome Array 2.0 platform. Three biological replicates of each cell type were used for the microarray analysis. A differential gene expression signature was defined by comparing microarray data from Huh7-R-A7 vs. Huh7-S cells. Signal intensities were analyzed by Affymetrix Expression Console software. Gene expression levels were RMA-normalized and log-transformed [44]. A filtering method based on the percentage of arrays (85%) below a noise cutoff of 6 (log2 scale) was applied to filter out low expression genes, and a linear model was employed to detect differentially expressed genes. In order to improve the estimates of variability and statistical tests for differential expression, a variance smoothing method with fully moderated t-statistic was employed for this study [45]. The significance level was adjusted for multiple hypothesis testing by controlling the mean number of false positives, and a threshold *p*-value < 0.0001 was maintained to determine statistical significance [46]. We averaged gene expression values for multiple probeset ID’s mapping to the same gene. A fold-change cutoff of >3 for up-regulated genes and <0.25 for down-regulated genes was imposed. Raw and normalized data were deposited in the Gene Expression Omnibus (GEO) database (accession: GSE94550).

#### 4.8.2. HCC Tumor Gene Expression Data

A systematic search of the Gene Expression Omnibus (GEO) database was conducted for datasets containing human HCC and normal liver tissue. We obtained raw (CEL files) gene expression data from six GEO microarray datasets: GSE14323 (GPL571), GSE14520 (GPL571), GSE14520 (GPL3921), GSE45267 (GPL570), GSE62232 (GPL570) and GSE6764 (GPL570). Signal intensity values were RMA-normalized and gene expression values were log-transformed. Differential gene expression analysis (tumor vs. normal) was conducted via the limma (Linear Models for Microarray Analysis) R package [47], which employs an empirical Bayes method to moderate the standard errors of estimated log-fold changes. We averaged gene expression values for multiple probeset ID’s mapping to the same gene. Benjamini–Hochberg False Discovery Rate (FDR) correction was applied to adjust for multiple hypotheses testing, and significance cutoff was set at adjusted *p* < 0.0001 [48]. Gene overlap Venn diagrams were generated using Venny 2.1.0 (available at https://bioinfogp.cnb.csic.es/tools/venny/index.html).

#### 4.8.3. RNA-Seq Analysis of Fostamatinib-Treated Huh7-R cells

RNA-Seq reads of DMSO- or fostamatinib-treated Huh7-R cells (n = 3 each) were first mapped to the human genome hg19 using Hierarchical Indexing for Spliced Alignment of Transcripts (HISAT) [49]. Raw read counts for each gene were quantified by using the featureCounts program [50]. Then RNA-seq counts were obtained by using GENCODE v.22 Gene Transfer Format (GTF) file as a transcript reference (GENCODE annotation). Genes with read counts below 5 for at least 2 samples out of 3 within each group were filtered out. Then the read counts were normalized with the TMM method [51]. To identify genes differentially expressed between samples, the limma R package was used to calculate *p*-values for group comparisons under a linear model [47]. The *p*-value cut-offs were determined by controlling the mean number of false positives [52]. Raw and normalized data were deposited in the Gene Expression Omnibus (GEO) database (accession: GSE113005).

### 4.9. Library of Integrated Network-Based Cellular Signatures (LINCS) Analyses

Connectivity mapping analyses were conducted via the Library of Integrated Network-based Cellular Signatures (LINCS) system (database version A2) using the web-based platform (http://www.lincscloud.org/). Connectivity scores were calculated using the weighted Kolgomorov-Smirnov (KS) statistic to rank predictions from the LINCS database, as previously described [12]. We selected LINCS compound perturbations tested exclusively in the HepG2 liver cancer cell line. Connectivity scores were averaged for individual LINCS compound perturbations tested at different concentrations and time points in the HepG2 cell line. Individual LINCS gene signatures were obtained from the Broad LINCS Cmap C3 Cloud Compute platform using the slice_slice_tool command.

### 4.10. Database Access: DrugBank, AACT, KEGG, Cell Miner, Beegle

We downloaded the “Approved” and “Investigational” external drug link files from the DrugBank database (v 4.5.0, released date 20 April 2016, available at https://go.drugbank.com/releases/4-5-0) and conducted searches in the DrugBank web interface to categorize drug approval status [53]. We downloaded the Aggregate Analysis of ClinicalTrials.gov (AACT) database, a publicly available resource produced by the Clinical Trials Transformation Initiative (CTTI), for drugs under clinical investigation (AACT accessed: 17 June 2016). We used KEGG DRUG database ID’s mapped to drugs in DrugBank to extract drug activity and target pathway information in KEGG Drug (KEGG DRUG accessed: 20 June 2016) [54]. Drug target genes were obtained from DrugBank and KEGG Drug databases, and assessed for association with HCC using the Beegle literature-mining tool [18]. Gene expression signatures for 18 HCC cell lines vs. a pool of 19 normal liver samples were obtained via the CellMinerHCC database [55].

### 4.11. Bioinformatics Analyses

Hierarchical clustering using the Euclidean distance of gene expression and drug connectivity scores was performed using the heatmap.2 function from the “gplots” R package. The nearest-template prediction (NTP) method was applied to normalized, log-transformed gene expression data to classify tumor samples as SR+ or SR− (FDR < 0.05) [56]. Gene mutation, copy-number and mRNA expression data for SR+/SR− liver HCC (LIHC) patients in the TCGA analysis were obtained via the cBioPortal (v 1.2.4) tool [57]. The drug target protein-protein interaction (PPI) network was generated using the STRING (v 10.0) database [58]. Only high confidence interaction scores (0.700 and above) from experiments, databases, neighborhood and gene fusion sources were included in the final network. Network analysis of the drug target PPI network was performed using Gephi 0.9.1 software [59], including algorithms to detect the following network features: degree, modularity class, eigencentrality and clustering coefficient. Enriched gene ontology functions in the sorafenib resistance gene signature were obtained via functional enrichment analysis using the ToppGene Suite [60]. Fostamatinib-treated HepG2 gene expression data from LINCS was analyzed via QIAGEN’s Ingenuity Pathway Analysis (IPA, QIAGEN Redwood City, CA, USA, www.qiagen.com/ingenuity).

### 4.12. Survival Analysis

Clinical data and RNAseq data (Level 3, v2, RSEM-normalized) from 377 liver HCC patients contained in the TCGA database were obtained via the Broad Institute Firebrowse tool (http://firebrowse.org/; TCGA data version 2016_01_28). Survival analysis comparing SR+ and SR− patients was conducted using Prism 7 software. Survival analysis of HCC patients for SYK based on gene expression data was conducted using the cBioPortal (v 1.2.4) tool for TCGA data [57]. Death was selected as the survival measure, and the median was chosen as the bifurcation point to define “high” vs. “low” gene expression. Survival curves were generated using the Kaplan–Meier method, and the log-rank test was used for statistical comparison.

### 4.13. Ethics Information for Publicly Available Datasets

Patient gene expression data were obtained from publicly available databases, including Gene Expression Omnibus (GEO) and The Cancer Genome Atlas (TCGA). We confirm that the publicly available data were collected with institutional approval and informed consent, and this information can be found in the original study publications: GSE14323 [61], GSE14520 [62], GSE45267 [63], GSE62232 [64], GSE6764 [65] and GSE26391 [16]. Data collection policies for the TCGA can be found at the following website: https://cancergenome.nih.gov/abouttcga/policies.

## 5. Conclusions

We show the feasibility of the drug repurposing workflow by validating novel drugs to use alone and in combination with sorafenib in HCC. Our analysis of publicly available gene expression datasets of sorafenib resistance models and HCC patient datasets to determine prioritized drug candidates may inform future additional validation studies. Future studies utilizing data from HCC patient samples with resistant disease will be needed as it becomes available. Importantly, future studies of novel HCC drugs must carefully consider safety and toxicity profiles, as one of the greatest barriers to clinical trial success for approval in this patient population is liver function [4].

## Figures and Tables

**Figure 1 cancers-12-02730-f001:**
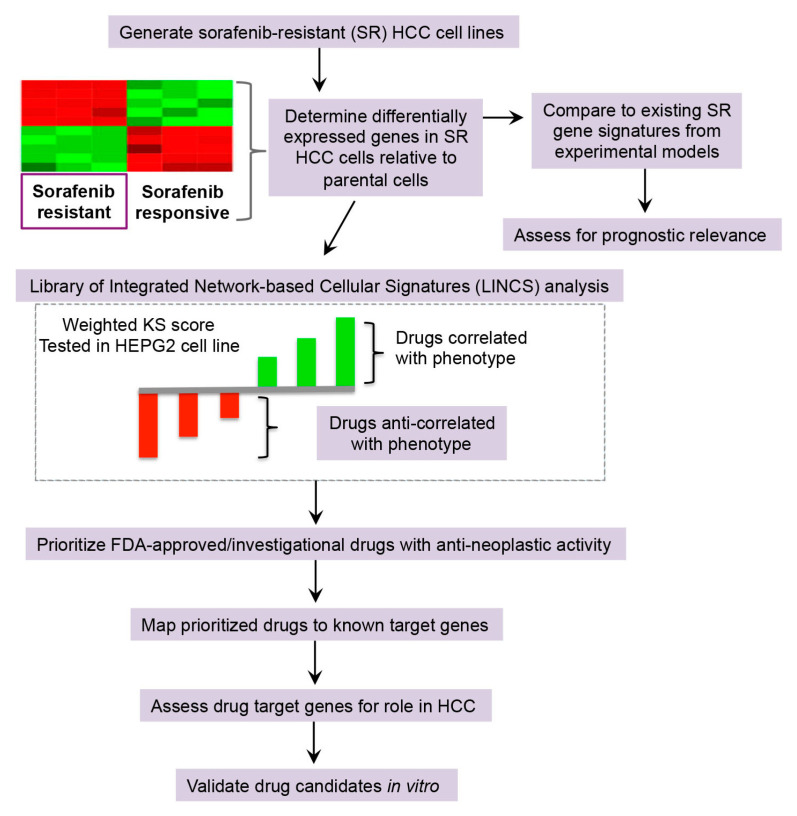
Overview of computational drug repurposing workflow and drug target network. Gene expression signatures of experimental models of hepatocellular carcinoma (HCC) sorafenib resistance were (i) assessed for prognostic significance, and (ii) queried against gene expression signatures characterizing drug perturbations in the HepG2 cell line contained in the Library of Integrated Network-based Cellular Signatures (LINCS) database. Connectivity scores were calculated by the rank-based, non-parametric weighted Kolgomorov-Smirnov (KS) statistic. Drugs with negative connectivity scores (i.e., anti-correlated) represent those that are hypothesized to reverse HCC sorafenib resistance gene signature. Drug candidates were further prioritized based on FDA approval/clinical investigation status, known anti-neoplastic activity and literature evidence for drug target genes associated with HCC. Two drugs were subsequently selected for in vitro validation.

**Figure 2 cancers-12-02730-f002:**
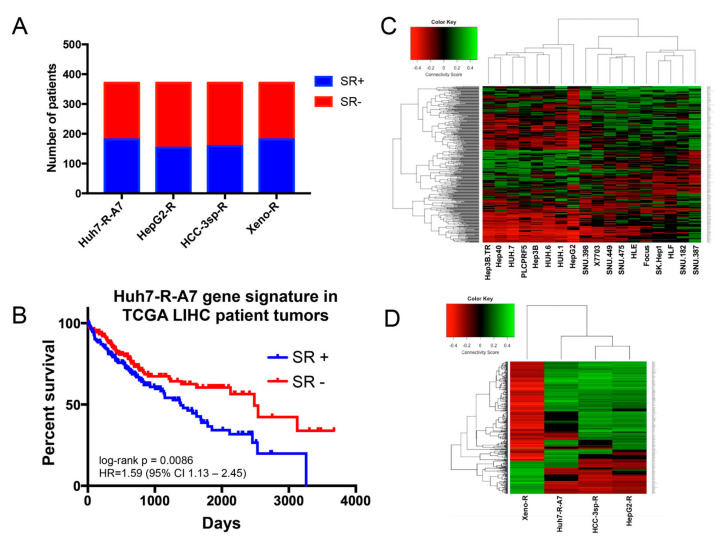
Sorafenib resistance gene signatures—prognostic significance and drug repurposing candidates. (**A**) Percentage of primary tumor samples in The Cancer Genome Atlas (TCGA) liver hepatocellular carcinoma (LIHC) dataset containing the four sorafenib resistance gene signatures (SR+). (**B**) Kaplan–Meier plot of overall survival of TCGA LIHC patients with primary tumors harboring the Huh7-R-A7 SR gene signature (n = 181 SR+) and those with primary tumors containing the inverse SR gene signature (n = 190 SR-). Heatmap visualization of hierarchical clustering analysis of connectivity scores for LINCS drugs derived from gene expression profiles of (**C**) HCC cell lines (n = 18) from the CellMiner database and (**D**) HCC sorafenib resistance (SR) experimental models.

**Figure 3 cancers-12-02730-f003:**
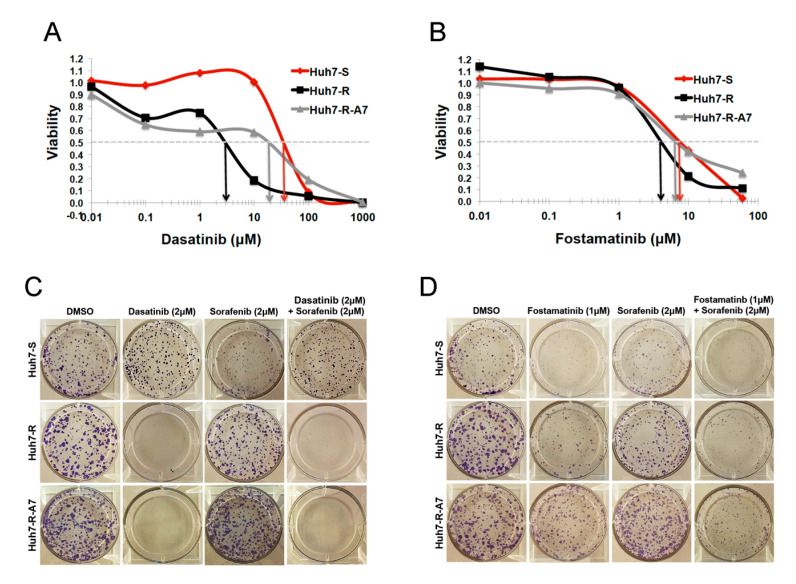
Validation of predicted drug candidates for action against sorafenib resistant HCC. Increased sensitivity of sorafenib-resistant Huh7 cells to dasatinib (**A**) and fostamatinib (**B**), as measured 48 h post-treatment using CellTiter-Glo viability assay. Growth of sorafenib-resistant Huh7 cells is inhibited by dasatinib (**C**) and fostamatinib (**D**), alone and in combination with sorafenib, as measured by colony formation assay 2 weeks post-treatment.

**Figure 4 cancers-12-02730-f004:**
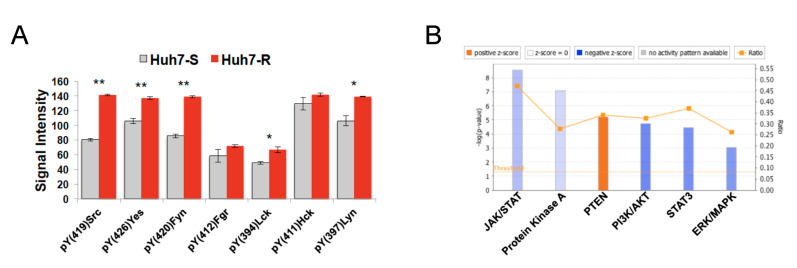
Mechanism of action of predicted drug candidates. (**A**) Enhanced phosphorylation of the Src family of kinases in sorafenib-resistant cells relative to sensitive cells, as measured by the Proteome Profiler Human Phospho-Kinase Array (* *p* < 0.05, ** *p* < 0.010, two-tailed t test) at 48 h. (**B**) Ingenuity pathway analysis of gene expression profile from fostamatinib-treated HepG2 cells reveals significantly altered pathways.

**Figure 5 cancers-12-02730-f005:**
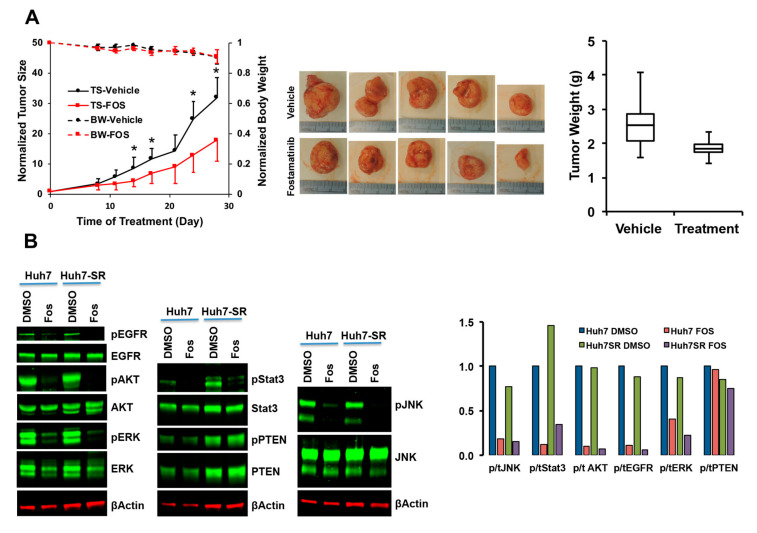
In vivo efficacy of fostamatinib and signaling pathway. (**A**) NSG mice bearing subcutaneous tumors of MHCCLM3 cells were treated with vehicle or fostamatinib (Fos) for 4 weeks. Body weights of tumor-bearing mice (BW) and size of the tumor (TS) during treatment, as well as pictures of tumors and tumor weights at the end of the treatment, are provided. (**B**) Western blot analysis of indicated proteins and quantification of phosphorylated protein normalized to total protein in Huh7 and Huh7-SR cells treated with vehicle (DMSO) or fostamatinib (Fos). Uncropped Western blots are shown in Appendix A. * Indicates a statistically significant difference between the two groups at *p* = 0.05.

**Figure 6 cancers-12-02730-f006:**
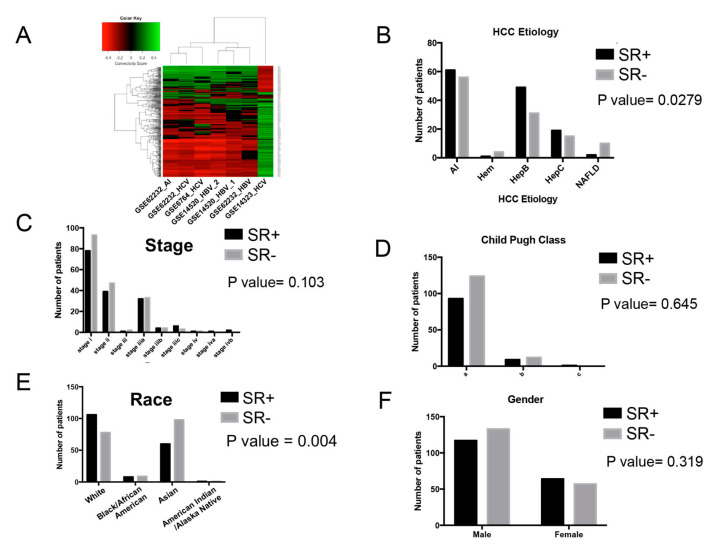
Impact of clinical and demographic factors on drug repurposing candidates and sorafenib resistance gene signature. (**A**) Heatmap visualization of hierarchical clustering analysis of connectivity scores for LINCS drugs derived from gene expression profiles of HCC patient tumor datasets from the GEO database representing distinct etiologies. (**B**) Proportions of SR+ and SR- patient primary HCC tumors in the TCGA LIHC dataset across five HCC etiologies (chi-square test). Proportions of HCC patients with primary SR+ and SR- tumors in the TCGA LIHC dataset across (**C**) stage based on TNM classification (chi-square test *p* value reported), (**D**) Child–Pugh class (chi-square test *p* value reported), (**E**) patient race (chi-square test *p* value reported) and (**F**) patient gender (Fisher exact test *p* value reported). Note the mean age at diagnosis for SR+ and SR− patients is 59.5 years and 59.4 years, respectively. Abbreviations: AI = alcohol-induced, HBV = hepatitis B virus, HCV = hepatitis C virus, NAFLD = non-alcoholic fatty liver disease, Hem = hemochromatosis.

**Table 1 cancers-12-02730-t001:** Description of HCC patient gene expression datasets and classification of HCC tumor vs. normal liver status by sorafenib resistance gene signatures.

HCC Patient Dataset		GSE14323	GSE14520_GPL571	GSE14520_GPL3921	GSE45267	GSE62232	GSE6764
**Number HCC tumor/normal liver samples**		n = 22 tumor/n = 19 normal	n = 38 tumor/n = 19 normal	n = 214 tumor/n = 220 normal	n = 31 tumor/n = 24 normal	n = 81 tumor/n = 10 normal	n = 35 tumor/n = 10 normal
Microarray platform		Affymetrix Human Genome U133A 2.0 Array	Affymetrix Human Genome U133A 2.0 Array	Affymetrix HT Human Genome U133A Array	Affymetrix Human Genome U133 Plus 2.0 Array	Affymetrix Human Genome U133 Plus 2.0 Array	Affymetrix Human Genome U133 Plus 2.0 Array
HCC etiology		HCV	HBV	HBV	n/a; HCC dx >40 yrs	AI, HBV, HCV	HCV
**Huh7-R-A7 (n = 368 genes)**	% SR+ samples	53.9%	55.8%	44.5%	50.6%	58.2%	53.30%
	P value	**<0.0001**	0.4225	**0.019**	0.5307	**0.0397**	**0.073**
	Sensitivity	0.56 (95% CI: 0.37–0.72)	0.53 (95% CI: 0.30–0.75)	0.47 (95% CI: 0.38–0.56)	0.43 (95% CI: 0.27–0.61)	0.42 (95% CI: 0.29–0.57)	0.61 (95% CI: 0.39–0.80)
	Specificity	1.0 (95% CI: 0.82–1.0)	0.67 (95% CI: 0.35–0.88)	0.70 (95% CI: 0.60–0.76)	0.69 (95% CI: 0.44–0.86)	1.0 (95% CI: 0.68–1.0)	0.86 (95% CI: 0.49–0.99)
**HepG2-R (n = 1147 genes)**	% SR+ samples	32.2%	20.9%	22.5%	24.1%	27.5%	30.7%
	P value	**0.018**	**0.0405**	**<0.0001**	**<0.0001**	**0.0212**	0.1139
	Sensitivity	0.42 (95% CI: 0.26–0.59)	0.54 (95% CI: 0.29–0.77)	0.53 (95% CI: 0.45–0.60)	0.57 (95% CI: 0.41–0.72)	0.42 (95% CI: 0.31–0.55)	0.68 (95% CI: 0.46–0.85)
	Specificity	0.93 (95% CI: 0.70–1.0)	0.88 (95% CI: 0.64–0.98)	0.90 (95% CI: 0.85–0.94)	0.97 (95% CI: 0.84–1.0)	1.0 (95% CI: 0.87–1.0)	0.67 (95% CI: 0.35–0.88)
**HCC-3sp-R (n = 847 genes)**	% SR+ samples	31.0%	23.3%	25.6%	31.0%	28.6%	26.7%
	P value	**0.0007**	0.0538	**<0.0001**	**0.0168**	**0.0443**	**0.0443**
	Sensitivity	0.45 (95% CI: 0.28–0.63)	0.50 (95% CI: 0.28–0.72)	0.46 (95% CI: 0.39–0.53)	0.49 (95% CI: 0.34–0.64)	0.40 (95% CI: 0.29–0.52)	0.57 (95%CI: 0.37–0.76)
	Specificity	1.0 (95% CI: 0.82–1.0)	0.87 (95% CI: 0.62–0.98)	0.83 (95% CI: 0.77–0.88)	0.79 (95% CI: 0.62–0.89)	1.0 (95% CI: 0.68–1.0)	0.86 (95% CI: 0.53–0.99)
**Xeno-R (n = 175 genes)**	% SR+ samples	25.2%	60.5%	50.1%	62.1%	40.7%	40.0%
	P value	0.0686	**<0.0001**	**<0.0001**	**<0.0001**	**0.008**	**0.0049**
	Sensitivity	0.33 (95% CI: 0.15–0.58)	0.82 (95% CI: 0.52–0.97)	0.86 (95% CI: 0.77–0.92)	0.86 (95% CI: 0.67–0.95)	0.6 (95% CI: 0.42–0.75)	0.93 (95% CI: 0.70–1.0)
	Specificity	0 (95% CI: 0–0.56)	1.0 (95% CI: 0.80–1.0)	0.99 (95% CI: 0.96–1.0)	1.0 (95% CI: 0.89–1.0)	1.0 (95% CI: 0.65–1.0)	0.80 (95% CI: 0.38–0.99)

For each of the six gene expression datasets from the Gene Expression Omnibus (GEO) database, the number of HCC tumor samples, normal liver samples, microarray platform type and HCC etiology for tumors are shown (AI= alcohol-induced, HBV= hepatitis B virus, HCV= hepatitis C virus). Fisher exact text P values (bold values indicating statistical significance at *p* < 0.05), sensitivity and specificity measures are shown for each of the four sorafenib resistance (SR) gene signatures across the six gene expression datasets to classify tumor vs. normal liver tissue statsus: Huh7-R-A7 (Huh7 cell line), HepG2-R (HepG2 cell line), HCC-3sp-R (short term culture of HCC patient), Xeno-R (xenograft of Huh7 cells). Bold numbers indicate statistical significance at *p* < 0.05.

**Table 2 cancers-12-02730-t002:** Prioritized LINCS drug predictions for reversing HCC sorafenib resistance using the Huh7-R-A7 gene signature.

Drug	Connectivity Score	Status	Target(s)	Drug Action(s)
**dasatinib**	−0.3073	Approved	ABL1,SRC,EPHA2,LCK,YES1,KIT, PDGFRB,STAT5B,ABL2,FYN	antineoplastic, kinase inhibitor
**enzalutamide**	−0.3686	Approved	AR	antineoplastic, antiandrogen, receptor antagonist
**paclitaxel**	−0.2423	Approved	TUBB1,BCL2,NR1I2,MAP4,MAP2,MAPT	antineoplastic, antimicrotubule
**palbociclib**	−0.2387	Approved	CDK4,CDK6	antineoplastic, kinase inhibitor
**pemetrexed**	−0.2830	Approved	TYMS,ATIC,DHFR,GART	antineoplastic, antimetabolite, antifolate
**toremifene**	−0.3114	Approved	ESR1	antiestrogen, antineoplastic, receptor antagonist
aminoglutethimide	−0.3121	Approved	CYP19A1,CYP11A1	adrenocortical suppressant, antineoplastic, aromatase inhibitor
anastrozole	−0.3002	Approved	CYP19A1	antineoplastic, aromatase inhibitor
nilotinib	−0.2677	Approved	ABL1,KIT	antineoplastic, kinase inhibitor
procarbazine	−0.2573	Approved	DNA	antineoplastic, alkylating
thiotepa	−0.3158	Approved	DNA	antineoplastic, alkylating
vemurafenib	−0.3455	Approved	BRAF	antineoplastic, kinase inhibitor
verteporfin	−0.2223	Approved	n/a	antineoplastic, photosensitizer
**brivanib**	−0.2896	Investigational	VEGFR2,FGFR1,FGFR2	antineoplastic, kinase inhibitor
**fostamatinib**	−0.2388	Investigational	SYK	antineoplastic, anti-inflammatory, kinase inhibitor
darinaparsin	−0.3833	Investigational	n/a	antineoplastic, organic arsenical
enzastaurin	−0.2693	Investigational	PRKCB	antineoplastic, kinase inhibitor
orteronel	−0.2719	Investigational	CYP17A1	antineoplastic, antiandrogen
quizartinib	−0.3080	Investigational	FLT3	antineoplastic, kinase inhibitor
tipifarnib	−0.4090	Investigational	FNTB	antineoplastic, farnesyltransferase inhibitor

LINCS drug predictions were prioritized if they had negative connectivity scores in the HEPG2 cell line against the query sorafenib-resistant HCC gene signatures, a known approval or investigational status from the DrugBank and/or ClinicalTrials.gov databases, and antineoplastic function described in the KEGG Drug and DrugBank databases. Drugs in bold font targeted genes known to play a role in HCC from a systematic analysis of the biomedical literature.

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
