# Peer review of "Transcriptomics-Based Drug Repurposing Approach Identifies Novel Drugs against Sorafenib-Resistant Hepatocellular Carcinoma"

_cancers, 2020, doi:10.3390/cancers12102730_

Round 1
Reviewer 1 Report
In the article authors have focus on two anti-neoplastic drugs Dasatinib and Fostamatinib (along with 18 other FDA-approved/investigational drugs), to identify drug effectiveness against sorafenib resistant Hepatocellular carcinoma cells. They have used connectivity mapping using transcriptomic gene signature from in vitro derived SR-HCC cells. Transcriptomics data (micro-array and RNASeq) was curated and processed from different sources.
Although, the method is not novel, it targets a specific cancer (HCC) and used comparatively updated drug database. Similar approach has been applied with a variation for many other studies, including HCC and other cancers.
The article work has been supported by computational analysis and well experimental results. Although, experimental sections look fine, I think more explanation need to be provided for computational data analysis design and processing steps. Some of them includes,
- How many samples/replicates were used for microarray DEG analysis?
- Since the GEO data was used from different sources (GSEs) and different platforms (GPLs), how the data was combined?
- Did authors check for possible batch effects? Does the sample clustered based on their GSEs? It would be better if they could provide sample clustering before and after any batch effect correction method that was applied.
- What were data filtering parameters used?
- For consistency authors should choose either p-val or FDR as limiting cut-off. It would be better to clearly mention the p-val cutoff for type each analysis and report whether it was adjusted for multiple testing.
- What could be explanation for different cut-off for up and down regulated genes?
Reported IPA pathways are selected based on p-val cutoff of 0.05. Were they also significant after BH-correction? Were these the top pathways?
Also, most recent article in their references is from 2016. If possible, author should provide more recent information in the introduction and used latest information available. e.g authors have used 2012 Globocon data to report 2017 predictions, whereas reports from 2018 is already available online.
Although not necessary, it would be better of author use the latest hg38 genome rather than old hg19 (released 2009) for RNAseq mapping.
Line 520: Table S1 and Video S1 are is missing.
Supplement Figure 5 is mis-typed as ‘Supplemental Table S5’. Also, explanation of color coding is missing in legend.
Author Response
We thank you for your time in reviewing the manuscript and providing feedback that has improved its quality. Please see attached for response to each comment.

Reviewer 2 Report
This study from Regan-Fendt and colleagues describes how new mechanisms of tumor resistance can be identified using publicly available transcriptomic databases originating from patients, pre-clinical models and cell line samples. More specifically, they focused their interest on hepatocellular carcinoma, exhibiting resistance to the chemotherapy of reference Sorafenib. Interestingly, after identifying several transcriptomic signatures from sorafenib-sensitive and resistant tumors, they were able to link molecular pathways to FDA-approved or investigated drugs. Therefore, they identified Dasatinib and Fostamatinib as potential anti-cancer agents with a new indication for sorafenib-resistant hepatocellular carcinoma.
I think the scientific rationale and overall interest of the study is really high, especially when considering the very short lifespan of patients diagnosed with HCC and the absence of potent therapies. I particularly liked the analysis pipeline used to find new targets from publicly available datasets.
However some issues with the manuscript have to imperatively addressed.
1) It could be an issue on my side, but I was unable to find the Tables S1, S2, S3, S4 and the Figure S4 (although I think the latter has been named Table S5 in the document that has been provided to me). Please check the document containing the supp material, I uploaded it to my review.
Although those tables are not critical to the manuscript's overall understanding, I would have liked to see the list of genes defining the transcriptomic signatures you were able to identify, for instance.
2) The preclinical study presented in Figure 5 is somehow underwhelming. First because except for the basic tumor growth follow-up, no further study was performed on those samples. For instance, why not trying to measure the phosphorylation states of the markers in Figure 5B in the xenografts samples instead of going back to the cell lines?
I am also curious to know what is the mechanism driving the tumor size reduction. Were the fostamatinib-treated tumors exhibiting necrotic areas, higher cell death count or reduced cell proliferation?
Also, why not opting for an orthotopic model instead of a subcutaneous xenograft?
Why no combined treatment with sorafenib was attempted despite the cumulative effects on tumor cell eradication presented in the Fig. 4? I understand that the MHCCLM3 has reduced sensitivity to sorafenib but so does the Huh7-R cells, yet you were able to kill nearly all colonies with the combined treatment.
3) There is no reference to the Proteome Profiler for phospho proteins kit / Kinase array (line 356) used to obtained the data in Fig. 4. Please add the information to the material and methods and if possible, add the full dot blots to the supplementary materials. How many times this experiment was repeated?
4) I am also surprised that you do not use phosphatase inhibitors (https://www.sigmaaldrich.com/catalog/product/roche/phossro?lang=fi®ion=FI) for the preparation of your protein samples. That might affect greatly the detection levels of the phosphorylated AKT, ERK and EGFR. Please comment on that.
Minor issues:
1) Please edit the figure 2, the font of the legends on panels C and D is very small, and zooming in reduces the quality. Because the supplementary tables are missing, I have no clue what are the genes listed on the left.
2) I personally feel that the last part of the results section dedicated to the clinical and demographic factors is a bit off topic and takes away the main message conveyed by the rest of the manuscript (the actual repurposing of fostama- and dasatinib). Also, isn't it more accurate to talk about ethnicity rather than race? I also think Native American is the correct terminology than Indian American, unless the dataset contains data about Indian immigrants to the US.

Author Response

(The authors gave the same response as above.)

Reviewer 3 Report
The manuscript by Kelly Regan-Fendt and coworkers provides a proof of concept evidence for a transcriptomic-based approach for the identification of novel potential drugs for sorafenib-resistant hepatocellular carcinoma (HCC) treatment. Moreover, the authors validate their results in in vitro and in vivo models of HCC.
The topic is interesting, the manuscript is well written and the experimental design appears robust. However, concerns are raised with regard to the lack of sufficient novelty.
Moreover, there are additional points that need to be addressed.
Main points:
- Please, provide more recent references about global cancer burden statistics referred to HCC (lines 41-42).
- In cell viability and colony formation assays the authors refer to 1% final DMSO concentration in media containing tested drugs. This percentage, if referred to the maximum vehicle DMSO concentration achieved in each well, is unacceptable in cell biology experiments performed to analyze the effects of a drug. Please, provide clarification about this concern.
- To evaluate expression of phosphoproteins in western blot analyses, one should employ both protease and phosphatase inhibitor cocktails in preparation of whole cell extracts. However, the authors missed such information.
- The authors throughout the manuscript refer to Supplementary Tables S1-S4 that they didn’t upload, thus I am not able to discuss them.
- The authors use terms as “synergistically” and “additively” (lines 345 and 353, respectively) in combination experiments, but they didn’t check these features by calculation of combination indexes, as assessed using CompuSyn software and Chou-Talalay equation (see references: Chou, T. C. Drug combination studies and their synergy quantification using the Chou–Talalay method. Cancer Res. 70, 440–446, 2010.; and, specifically, Florio et al, Effects of dichloroacetate as single agent or in combination with GW6471 and metformin in paraganglioma cells. Sci Rep. 8, 13610, 2018, for evaluation of the effects of single, or combined treatments on cancer cell clonogenic capacity). Accordingly, calculate the combination indexes for combination experiments depict in Figure 3C-D. Moreover, provide quantitative analyses for colony formation assays depict in the same Figure.
- In in vivo experiments, there is a lack of information about toxicity of fostamatinib in the animals.
Minor points:
- There are a few typos throughout the text (line 258, “and” is missed; line 360, please correct micromolar and hrs abbreviation; in Figure 3C, please correct “dasamatinib”; “Supplemental Table S5” should be “Supplemental Figure S5”).
- In Figure 1, please remove “Figure 1” label from the image.
- Please, improve resolution of Figures 2C, 6A and 6E (this latter panel should be better repositioned).
- Please, provide statistical analyses for data in Figure 5A (box plots) and for densitometric quantification in Figure 5B.
- Information about FOS concentration employed in western blot analyses are missing, both in Figure 5 and in Materials and Methods.
- Please, provide additional labels for HCC cell lines tested in Supplemental Figure S4.
Author Response

(The authors gave the same response as above.)

Reviewer 4 Report
The current manuscript entitled, “Transcriptomics-based drug repurposing approach identifies novel drugs against sorafenib-resistant hepatocellular carcinoma” by Regan-Fendt et al. is an excellent work to identify the newer more effective treatment for hepatocellular carcinoma for which sorafenib is the only approved first-line therapy and majority of patients become resistant to sorafenib. The authors collectively used information available in public domain and supported with in vitro and in vivo work in their laboratory to provide proof of concept data for Dasatinib and Fostamatinib. The manuscript in its current format has answered very critical questions with controlled experiments and novel results but there are few suggestions which will help readers use useful information to understand and appreciate the work.
- Authors have used Huh7-R or A7 clone for most of the experimentation in the manuscript. Author should explain or provide rationale/information on why they used the MHCCLM3 cell line to generate the tumor model in mice? The current rationale that MHCCLM3 cells are inherently reduced sensitivity to sorafenib is provided but why did the author not use Huh7-R-A7 cells or any other clone to generate the animal model to show fostamatinib effect?
- How did Author select concentration of fostamatinib for in vivo study?
- Did the author try using Dasatinib and Fostamatinib combination in Sorafenib resistant cell lines? The concentration of 2 uM used completely inhibited proliferation of Sorafenib resistant cells and Dasatinib seems to be acting on a different pathway than Fostamatinib so what happens if authors use nanomolar concentration of Dasatinib in combination with Fostamatinib? Is there a synergistic effect or any other observation?
- Simplify guilt-by-association approaches in introduction or add meaning/explain. Readers should not have to go and search for the context from introduction. This is an optional suggestion.
Author Response

(The authors gave the same response as above.)

Round 2
Reviewer 1 Report
Authors have answered most of the question and implemented the suggestions.
However, for point 4, authors responded, “All filtering parameters used are already described in the methods (see lines 168 & 198)”.
There is no parameter related information in line 168 “...down-regulated genes was imposed. Raw and normalized data were deposited in the Gene Expression...” and 198 “...Connectivity scores were averaged for individual LINCS compound perturbations tested at different…”. It would be better if authors could define percentage of arrays , noise cutoff, etc.
In table 1, is specificity=0 for Xeno R (GSE14323)? Also, except for Xeno-R, sensitivity seems less for all cases. Can authors provide information on prevalence adjusted PPV and NPV?
Reviewer 2 Report
Thanks to the authors for editing the manuscript and providing us with the missing figures.
I still have some disagreement on the authors' answers to my comments that I am gonna list below. Although they do not justify a rejection of the manuscript, I encourage the authors to take them in account while finalizing the submission.
1) On an ethical and scientific point of view, using laboratory animals simply for a "proof-of-concept" study seems like a waste of ressources and uneccessary use of live animals. If the authors wanted to deliver a proof-of-concept, in vitro experiments were just enough.
2) Subcutaneous models are outdated, the frequency of use in the litterature is not a a solid scientific reason to justify using it.
3) As for my concerns about the race/ethnicity denomination, I don't think Oprah Winfrey's community managers are a reliable source to determine the right scientific term. I would encourage the authors to extend their bibliography research, e.g.
Sober, Elliott (2000). Philosophy of biology (2nd ed.). Boulder, CO: Westview Press. pp. 148–151. ISBN 978-0813391267.
Keita, S O Y; Kittles, R A; Royal, C D M; Bonney, G E; Furbert-Harris, P; Dunston, G M; Rotimi, C N (2004). "Conceptualizing human variation". Nature Genetics. 36 (11s): S17–S20. doi:10.1038/ng1455. PMID 15507998. Many terms requiring definition for use describe demographic population groups better than the term 'race' because they invite examination of the criteria for classification.
Reviewer 3 Report
The manuscript in the present revised version is now suitable for publication.
Author Response
Thank you.